# Reconstruction of Vascular and Urologic Tubular Grafts by Tissue Engineering

**Christophe Caneparo** [1], **Stéphane Chabaud** [1] and **Stéphane Bolduc** [1,2,*]

1   Centre de Recherche en Organogenèse Expérimentale de l'Université Laval/LOEX, Centre de Recherche du CHU de Québec-Université Laval, Axe Médecine Régénératrice, Québec City, QC G1J 1Z4, Canada; christophe.caneparo@crchudequebec.ulaval.ca (C.C.); stephane.chabaud@crchudequebec.ulaval.ca (S.C.)
2   Department of Surgery, Faculty of Medicine, Université Laval, Québec City, QC G1V 0A6, Canada
*   Correspondence: stephane.bolduc@fmed.ulaval.ca

**Abstract:** Tissue engineering is one of the most promising scientific breakthroughs of the late 20th century. Its objective is to produce in vitro tissues or organs to repair and replace damaged ones using various techniques, biomaterials, and cells. Tissue engineering emerged to substitute the use of native autologous tissues, whose quantities are sometimes insufficient to correct the most severe pathologies. Indeed, the patient's health status, regulations, or fibrotic scars at the site of the initial biopsy limit their availability, especially to treat recurrence. This new technology relies on the use of biomaterials to create scaffolds on which the patient's cells can be seeded. This review focuses on the reconstruction, by tissue engineering, of two types of tissue with tubular structures: vascular and urological grafts. The emphasis is on self-assembly methods which allow the production of tissue/organ substitute without the use of exogenous material, with the patient's cells producing their own scaffold. These continuously improved techniques, which allow rapid graft integration without immune rejection in the treatment of severely burned patients, give hope that similar results will be observed in the vascular and urological fields.

**Keywords:** tissue engineering; biomaterials; self-assembly; blood vessel; ureter; urethra

## 1. Introduction

The human body is a complex multicellular organism. Cooperation between organs and between cells inside these organs is crucial to ensuring the survival of the whole. Many of these vital functions of the organism are carried out by tubular structures abundant in the body. For example, nutrients' absorption through the esophagus, processing and elimination of the waste by the gastrointestinal conduits for the non-soluble part or the urinary system for the soluble ones; bronchi for the oxygen; blood vessels for the distribution of processed nutrients and oxygen and the evacuation of metabolic waste including carbon dioxide. These tubes have various sizes, ranging from the smallest, such as capillaries or small ducts, to the larger ones such as the esophagus, intestine, vagina, nerve conduits, prominent veins and arteries, as well as the intermediate, such as ureters and urethra, small-size vessels and ducts. Most systems combine different sizes such as vascular system, breast, prostate, bronchi, and testes.

These tubes can be seen as the roads connecting specialized facilities. Their function is essential in ensuring all the organs are supplied with food and raw materials and that the waste is evacuated. Their engorgement or occlusion can have catastrophic consequences. When medical actions are required, traditional surgeries can suffer from the limited availability of convenient tissues: alternative solutions are required. Among them, tissue engineering (TE) has developed considerably in recent decades and provides promising avenues. In this review, we will focus on vascular and urologic TE, emphasizing a technique to produce tissues without exogenous materials called the "self-assembly method", developed in the 1990s by Dr. François A. Auger at LOEX (Quebec, QC, Canada).

## 2. Tissue-Engineered Vascular Graft

In the vascular system, the arteries carry the oxygenated blood from the heart to the organs. Capillaries irrigate these organs' cells, and the veins return the hypoxic blood to the heart for a new cycle of circulation. To reconstruct a functional tissue, it is essential to understand how it works and what the role of each part is.

### 2.1. Anatomy of Blood Vessels

This review will only discuss arteries and veins, excluding capillaries, which have a distinct anatomy. Arteries (Figure 1a) and veins (Figure 1b) have similar trilamellar anatomy [1].

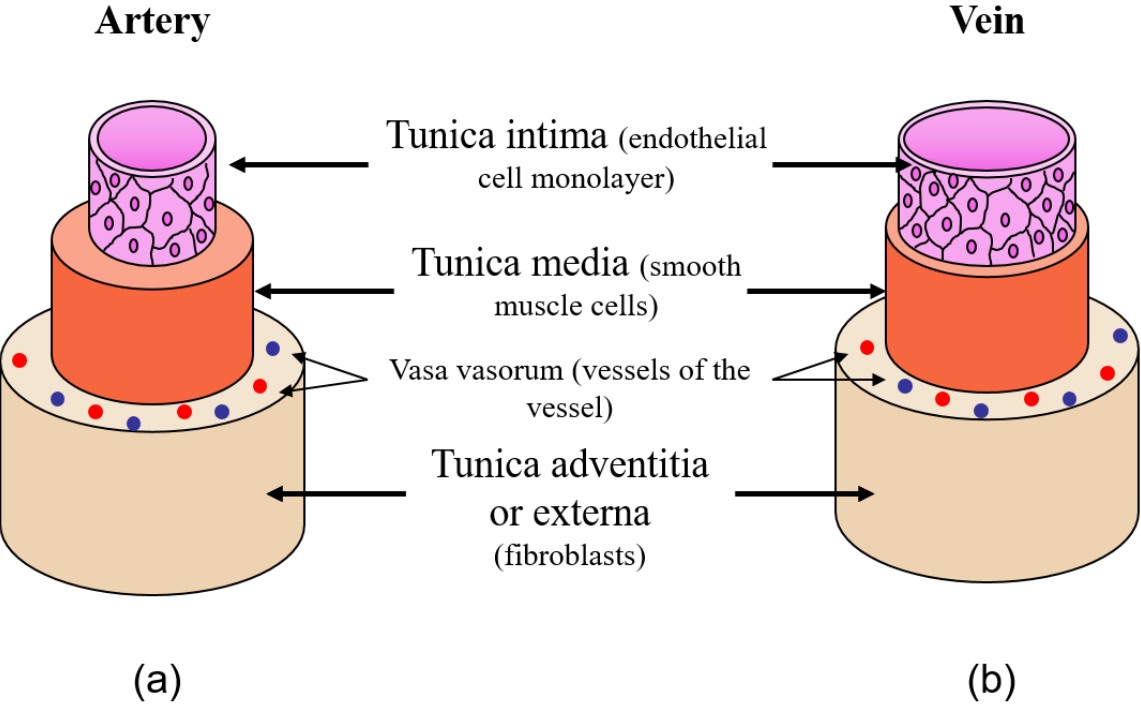

**Figure 1.** Schematized anatomy of vascular conduits. A schema of the arteries' anatomy is presented on (**a**), and the anatomy of veins is presented on (**b**).

The features required for blood vessels are the barrier function that strictly limit permeability, the ability to respond to signals by modifying the tone of the vessels (contractility), and the maintenance of mechanical properties to avoid non-elastic deformations, which can otherwise result in an aneurysm. From the internal lumen to the outer surface, the three layers are: The tunica intima (TI), which consists of a monolayer of endothelial cells on a basal lamina. TI prevents the thrombogenic response by forming a barrier separating blood and underlying layers. It also has a sensing function to regulate the tone of the vessel by secreting mediators. The tunica media (TM) is formed of smooth muscle cells (SMC) in an extracellular matrix (ECM) rich in collagen, elastin and proteoglycans. TM, by contracting and relaxing, is responsible for the tone of the vessel. Finally, the tunica adventitia (TA), or tunica externa, is composed of fibroblasts embedded in ECM to ensure the vessel's mechanical resistance. The vasa vasorum, which originates mainly from the outer layer, supplies the nutrients used to feed the vessel's cells and is essential when the vessel's thickness exceeds the nutrient/oxygen diffusion limit. The exact organization of the vasa vasorum depends on the size of the blood vessels [2]. TM also supports the innervation of the blood vessel. Elastic properties are provided by an elastin network which is mainly present at the individual layers' interface: the internal and external elastic lamina. Several differences exist between these two types of vessel. With a similar overall diameter, the lumen's diameter is more significant in the veins, to passively carry the blood to the heart.

In contrast, the TM is thicker in the arteries, which need to react to their environment and the messages of the body by contracting, narrowing the lumen, increasing blood pressure, or relaxing to reopen it. Therefore, a TE blood vessel needs to be biocompatible, have good mechanical properties and compliance, be suturable, be durable, have prolonged patency, and be anti-thrombogenic and vasoactive [3]. Off-the-shelf and inexpensive characteristics are also significant advantages.

### 2.2. Cardiovascular Diseases

From the World Health Organization, cardiovascular diseases (CVD) are the number-one killer, with 18 million death in 2017. CVD mortality rates are exceptionally high in developed countries, and CVDs take the form of an epidemic with dramatic consequences for human health [4,5]. At present, CVD is constantly and rapidly increasing in developing countries, where the majority of deaths occur [6]. Cardiovascular diseases regroup various pathologies affecting the circulatory system. Three of them are discussed below: atherosclerosis, aneurysm and consequences of chronic kidney disease.

The most common result of the global adoption of the Western lifestyle, unsuited to human genetics is atherosclerosis, which is characterized by a narrowing of the lumen of the blood vessels. This can lead to coronary artery disease, myocardial infarction, stroke and peripheral artery disease [7]. Atherosclerosis combines metabolic disorder with increasing Low-Density Lipoprotein (LDL) concentrations in the blood, and chronic inflammation. Atheroma plaques are formed by lipid deposition and the accumulation of infiltrated immune and smooth muscle cells [8,9]. The rupture of the plaque can lead to stenosis, and this occlusion blocks the blood flow.

Aneurysmal disease is characterized by the presence of an aneurysm, a bulge or ballooning in a weakened blood vessel, especially when the mechanical properties of the blood vessel become too low to support the internal pressure [10]. The progressive dilatation of the vessel can lead to rupture. Aneurysmal disease's overall death rate is 2.5 in every 100,000 cases [11].

Indirectly related to CVDs, chronic kidney disease and the kidneys' failure to appropriately filter the blood to evacuate waste present in the blood requires hemodialysis. This process weakens blood vessels by the vascular access used to connect the circulatory system to the dialysis machine, leading to a high morbidity and mortality rate. The number of treated patients reached 2.6 million people in 2010 and is expected to be 5.4 million in 2030.

### 2.3. Current Therapeutic Options

These clinical problems require either replacement/bypass or repair of the damaged blood vessel section.

To remove the occlusion caused by the atheroma plaque and re-establish a roughly normal blood flow, balloon angioplasty with or without stenting is a tempting solution. It consists of the insertion of an inflatable balloon-containing catheter into the stenotic zone. The balloon is then inflated to press the plaque and widen the lumen. Due to the weakening of the vascular wall, a stent can be placed to maintain the opening of the vessel [12,13]. Nevertheless, sometimes this procedure is not sufficient, or the problem recurs, mainly because the cause of the disease is not treated, only its consequence. Then, patients must be treated using more invasive methods, such as surgical procedures.

When a surgical procedure is required to bypass atherosclerotic lesions, autograft, mainly using the saphenous vein, and eventually allograft and xenograft, can be used. The two latter options are not the first choice due to, among other reasons, the possibility of contamination by various pathogens [14]. Nevertheless, the autograft using the saphenous vein is limited to the availability of adequate tissue, mainly due to the health status of the patient [15], or when surgeries need to be repeated [16]. The failure at 10–15 years of coronary artery bypass grafting (CABG) using autologous saphenous vein is as high as half of the patients [17,18].

Synthetic prosthetic grafts made from polytetrafluoroethylene (PTFE/Teflon, or extruded PTFE/Gore-Tex, a fluorocarbon polymer), polyethylene terephthalate (PET/Dacron, a polyester material), or polyurethane (PU) has been developed and became the gold-standard of surgical treatments to repair large-diameter vessels, i.e., more than 6 mm of internal diameter. Their poor elasticity, compliance and anti-thrombogenic properties constitute fewer defects [19,20]. For small-caliber vessels, clinical results show unsatisfactory outcomes, notably thrombosis and intimal hyperplasia [21,22]. Therefore, the use of saphenous vein remains the gold standard.

New therapeutic options must be quickly developed to answer the clinical needs, which are only partially fulfilled by the current gold-standard techniques, especially for small-diameter blood vessels.

*2.4. Advanced Therapeutic Options for the Reconstruction of Small-Diameter Blood Vessels: Tissue Engineering*

In 1993, Langer and Vacanti published a paper in *Science* entitled "Tissue Engineering." Even if this was not the birth of TE, which already existed in several forms, it is the first essay to conceptualize this notion [23]. From this date, TE considerably developed to offer new and innovative solutions to old problems. Tissue engineering consists of the in vitro reconstruction of tissues or organs for replacement/repair and 3D-models for fundamental research. Synthetic or natural biomaterials, including decellularized organs as scaffolds, seeded or not with cells, were used. Various techniques have been set-up to create the scaffold, such as casting, electrospinning or bioprinting. Other methods, like the self-assembly technique, which does not require any exogenous biomaterial, were also developed. The aims of vascular TE are, from a long-term perspective, to recreate an efficient barrier to blood, including at the anastomosis site, and to avoid thrombosis. This should maintain a confluent endothelium forming the TI. Therefore, it aims to sustain blood pressure, obtain adequate TA mechanical properties, and respond to stimuli to adapt the lumen-opening for blood flow and blood pressure with differentiated smooth muscle.

In the following sections, synthetic, such as poly($\varepsilon$-caprolactone) (PCL), polydioxanone (PDO), polyglycolic acid (PGA), polyglycerol-sebacate (PGS) or polylactic acid (PLA or Vicryl), and natural biomaterials such as collagen, elastin, fibrin, hyaluronan and silk, but also decellularized matrices, will be described (Table 1). A combination of synthetic and natural scaffold is also described. Two excellent recent reviews have been published on this subject [3,24], so we will only summarize these biomaterials. Afterwards, the self-assembly technique, also known as the cell-derived matrix technique, and its derivatives will be detailed.

2.4.1. Synthetic Biomaterials

Synthetic biomaterials present many advantages, such as their low cost and ability to be highly tunable and take on the desired shape. They also have some drawbacks, such as their differences in composition and organization compared to the natural ECM, which can impact the adhesion, proliferation and differentiation of cells. Nevertheless, significant progress has been made with the functionalization of synthetic biomaterials in recent decades.

To repair the blood vessels, an attractive strategy is to combine the advantages of the stent without its drawback, using biodegradable/bioresorbable materials. The biomaterial can sustain the blood flow after implantation and is gradually replaced by the patient's tissue, until a completely new section of the autologous blood vessel is functional. Because bioresorbable polymers are entirely dissolved in the body, they are more attractive than biodegradable ones, for which fragments persist after their degradation. The degradation/resorption rate is critical because the biomaterial should disappear in the same timeframe the native ECM uses to strengthen the neotissue.

PCL degrades in about two years, with a change in its mechanical properties with degradation [25]. Cast PCL seemed to have fewer attractive mechanical properties than electrospun PCL, producing a scaffold with properties close to the ones of the native tis-

sue [26]. PCL seemed to be able to be cellularized by the host, especially endothelialization, which avoids thrombogenic reaction [25]. PDO degrades slowly, and endothelialization of the lumen of PDO tubes has been observed. PDO is also used for airway and gastrointestinal TE applications. Its mechanical properties are similar to the native ECM [27]. PGA degrades faster than PLA, and both produce lactate during their degradation [28]. Even if lactate is a natural metabolite, its role in promoting tumor by modifying the microenvironment renders its use questionable [29]. PGA seems to degrade quickly, within two months, with a loss of mechanical properties [30]. This rapid degradation rate might not be sufficient for the host cell's colonization of the graft. PLA presents a longer degradation rate and could be found in the body years after implantation [31]. PGS is an elastomeric polymer with a 1- to 2-month degradation period in subcutaneous studies [32]. PGS can quickly be colonized by the host's cells, especially endothelial cells and SMC [33]. The latter is essential for the remodeling of the graft. PGS also enhanced elastin expression with increased mechanical properties [34].

Other well-known synthetic biomaterials, such as poly(lactic-co-glycolic) acid (PLGA) and poly-l-lactic acid (PLLA), could be used [24]. Various techniques can modify the biodegradation, mechanical features and biological properties of these various synthetic biomaterials. For example, the degradation rate can be modulated by the copolymerization of various synthetic biomaterials. The migration and proliferation of cells and differentiation can be increased by coating the material with peptides, RGD or derivatives, or proteins. In the same way, thrombogenic activities can be reduced. A combination of natural biomaterials in hybrid scaffolds will be described below. Surface modifications, including plasma surface treatment, can also be made by altering the properties of the material surface physically or chemically by modifying parameters such as its hydrophilicity.

2.4.2. Natural Biomaterials

Contrary to biosynthetic materials, natural biomaterials used as scaffolds, especially those coming from ECM, provide more convenient environments for the migration, proliferation and differentiation of the cells when used under appropriate conditions. Nevertheless, these biomaterials generally have weaker mechanical properties. Indeed, the mechanical properties of ECM are not only due to the molecules of which it is composed but also, and more importantly, the organization of these molecules in the matrix.

Collagens are the most abundant proteins of the human body, comprising approximately 30 percent of the body's total dry weight. Collagens are a significant component of connective tissues where they provide mechanical resistance. Type-I collagen is the most expressed protein of this 29-member family. In the blood vessels, it can be found in TA, produced by fibroblasts, and in smaller amounts in TM, produced by secretory SMC. Collagen has commonly been used in TE since the experiment by Weinberg and Bell [35]. Despite its very appealing cell-friendly properties [36], its thrombogenic properties [37] require the formation of a barrier between the biomaterial and the blood, such as an association with heparin [38] or endothelial cells' pre-seeding [39]. Collagen hydrogels, which do not present the organization of the collagen fibers in native ECM, present weak mechanical properties [40] that can be increased by modifying the fibers' alignment by the use of bioreactor or cyclic strains [41,42].

Recently, studies have demonstrated, using collagen as a biomaterial, modular units that can form tubular structures [43] and self-expandable tubes [44]. Elastin is also a major component of blood vessels, and the elastin network confers them with the good elasticity required to resist the blood pressure. One of the challenges with elastin lies in obtaining a network between the fibers. When artificial crosslinkers are used, they need to mimic the elastin network to preserve its mechanical properties adequately. It also seems logical to combine elastin with collagen to form scaffolds [45,46]; nevertheless, elastin can also be combined with synthetic biomaterials [47,48]. Fibrin is a derivative of fibrinogen, a protein cleaved by thrombin during the first steps of wound-healing and hemostasis, to form a clot

where cells can migrate, proliferate and secrete ECM elements [49,50]. Fibrin is also known for allowing capillary growth to nourish highly metabolic cells during tissue repair [51].

Nevertheless, fibrin has very low mechanical properties. It must be remodeled quickly to be replaced by organized ECM with appropriate features. To speed-up the process, cells can be embedded in fibrin scaffolds. Hybrid collagen/fibrin scaffolds are also used [52,53]. Hyaluronan is an abundant glycosaminoglycan whose mechanical properties can be tuned using several methods [54,55]. Usable by itself, it can also be combined with collagen [56]. Silk is one of the most surprising natural biomaterials used in TE [57–60]. Silk yarns come from an insect cocoon [61] or spider [62], providing elasticity and glue-like properties [63]. Biocompatible and non-thrombogenic, with excellent mechanical properties, silk could be a promising biomaterial to reconstruct vascular substitutes [63–67].

### 2.4.3. Hybrid Constructs

As mentioned before, a combination of synthetic and natural biomaterials were used to engineered vascular graft [48,68]. The idea is to benefit from the mechanical properties and the synthetic biomaterials' versatility while preserving the "cell-friendly" characteristics of the natural biomaterials; for example, three successive layers of PCL, collagen and PLLA [69], or a mix of PCL and collagen [70]. Such a scaffold was grafted in sheep and followed for six months without any stenosis, implying that collagen was reinforced by PCL, and PCL was functionalized by collagen [71]. The research is encouraging, and further studies could optimize such constructs in the future.

### 2.4.4. Decellularized Tissues

Among other promising biomaterials for the reconstruction of the small-diameter blood vessel by TE, decellularized tissues become increasingly attractive with new protocols of decellularization [72]. Indeed, what material could be better than the native one? Decellularizing blood vessels to remove antigenic molecules responsible for immunological response and potential pathogens before recellularization with the patient's cells allows the production of tissue similar to autograft, or even better, if the disease's causal effect is also corrected. The decellularization of tissue can be obtained by successive hypo/hypertonic shocks, by itself or combined with the addition of detergents, such as sodium-dodecyl-sulfate (SDS), sodium deoxycholate (SD), Triton X-100 or -200, 3-[(3-cholamidopropyl)dimethylammonio]-1-propane sulfonate (CHAPS), to destroy the cell membrane, and enzymes, to remove proteins and nucleic acids [73]. The tissue can originate from a human cadaver or be obtained from animals. Nevertheless, several problems slow down the use of such vascular prostheses: ethical problems in cadavers and various regulatory issues such as the risk of immune reaction in the case of ineffective/incomplete decellularization. However, the main problem is the ability to maintain the ECM architecture that provides good mechanical and biological characteristics to the graft. In the past, ECM was largely disorganized or modified by the decellularization process, and lost its mechanical properties, resistance, elasticity and cell differentiation potential. An increasing number of protocols have emerged to reduce these problems, but more studies must obtain a clear view for this option [74]. Several recent studies have also been published using various sources of tissue [75–78].

### 2.4.5. Techniques to Produce Reconstructed Vascular Graft

Using these various biomaterials, several approaches have been developed to reconstruct a TE vascular graft [3,24,79]. We will briefly describe electrospinning (Figure 2a), casting in a mold (Figure 2b) and bioprinting (Figure 2c).

### Electrospinning

Electrospinning is one of the most used techniques in TE. Its principle is straightforward: biomaterials are transformed into fibers projected in a high-voltage electrical field against a rotating cylindrical mandrel, creating a tubular structure [80]. The engineer

can control various parameters to obtain the expected results, such as size, porosity, and mechanical properties [31,81]. Electrospinning is mainly used with synthetic biomaterials and can also be performed using natural biomaterials or form hybrid tubes. A complete review on electrospinning for vascular tissue engineering is available [82]. Several studies have been published recently, indicating that electrospun biomaterials can be helpful in the reconstruction of TE blood vessels [83–88].

**Table 1.** Strengths and weaknesses of the biomaterials used in vascular reconstruction.

| Biomaterial | Strengths | Weaknesses | Ref. |
|---|---|---|---|
| PCL | Two-year biodegradation rate allowing cell colonization, good biocompatibility. | Currently limited to electrospinning | [25,26] |
| PDO | Slow biodegradation rate, biocompatibility. | Safety under investigation | [27] |
| PLA | Slow degradation rate (years) | Lactate as a metabolite | [28,31] |
| PGA | Degradation rate can be tuned by PLA association | Lactate as a metabolite. Fast degradation rate (2 months) can limit cell colonization. | [28,30] |
| PGS | Enhance elastin expression, good biocompatibility | Fast degradation rate (1–2 months). | [32–34] |
| Collagen | Highly biocompatible | Low mechanical properties, thrombogenic | [35–37,40] |
| Elastin | Component of the blood vessel | Need to be combined with other biomaterials | [45–48] |
| Fibrin | Biocompatibility, high angiogenesis, degrade like natural fibrin clot | Very low mechanical properties | [49–51] |
| Hyaluronan | Tunable mechanical properties. Good biocompatibility. | Produced in large quantities by microbial fermentation | [54,55] |
| Silk | Very low degradation rate. Good mechanical properties and biocompatibility. | Very low degradation rate. | [57–67] |
| Acellular matrix | Best scaffold for cells when composition/organization of ECM is kept. | Mechanical properties (depending on the protocol used), DNA and antigen residues. | [72] |

Casting

Cast is probably the most straightforward technique, which is compatible with virtually all biomaterials [79,89–91]. It is done by casting the biomaterial into a mold. Sequential castings can take place with various mandrel sizes to obtain different layers. Centrifuge force can also be used to coat the mold with biomaterial. Solidification of the material can be obtained by gelation, crosslink, evaporation/lyophilization, etc. As previously mentioned, the mechanical properties obtained from a specific material did not reach the expected values. Fusion between different layers must also be optimized to avoid delamination.

Bioprinting

Three-dimensional printing is widely used in engineering, but living tissue is slightly different from inorganic structures. This technique has recently been transferred to TE, primarily vascular TE [92,93]. One of the challenges of this technique lies in retaining viable cells into the printed bioink due to heat and mechanical load. Bioprinting shows a high degree of customization and control because the tissue elements can be deposited at their exact desired position [94–98]. Various biomaterials, especially the natural ones, such as collagen or fibrin, can enter the composition of the bioink [24,99]. In some protocols, cells can be seeded inside the material before being printed [100], or cell aggregates [101,102] can serve as the bioink. Nozzle-based 3D printers and laser-derived stereolithographic bioprinters are currently used and allow labor-intensive use, flexibility and versatility. Nevertheless, it is unclear questionable if the structure of the printed tissue will maintain, over the long term, the same coherence as obtained with the other techniques. Recently, several studies have been done using bioprinting to produce TE blood vessels [103,104].

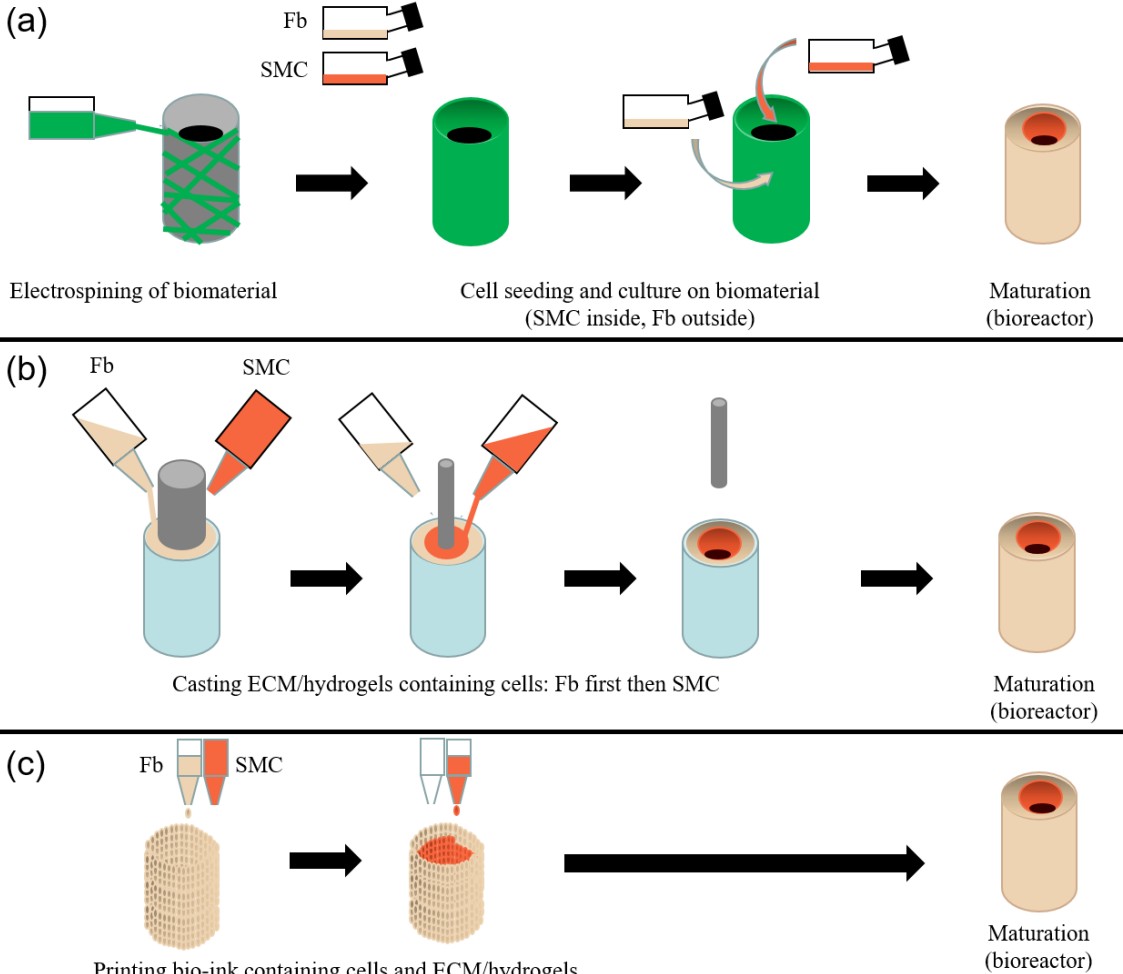

**Figure 2.** Principal techniques to produce vascular tubular structures. (**a**): Electrospinning; (**b**): Casting in a mold; (**c**): Bioprinting.

### 2.4.6. Tubular Structures Reconstructed by Self-Assembly

Scurvy has been known since ancient times, with the first mention of it in 1550 BC [105]. Clinical manifestations include bleeding, poor wound healing, and leg and gum disease. Fatalities can be caused by infection or bleeding. Complete deficiency in ascorbate can be reached after 4 to 12 weeks when no vitamin C is provided in the diet. With the development of European powers' colonial empires since the late 16th century, long voyages became common, and scurvy ravaged the crews. In 1617, John Woodall, a surgeon in the British East India Company's navy, recommended adding citrus and oranges to the diet of sailors for long expeditions. Despite John Lind, a surgeon of the Royal Navy aboard HMS Salisbury, leading the first scientific experiments in 1747 demonstrating that the supplementation of seamen's diet with citrus reversed the scurvy, citrus rations were not introduced before 1795 [106,107]. The demonstration that scurvy was a disease of connective tissue, and the discovery of Vitamin C's role, also called ascorbate, in the healing of the patients, as well as the mechanisms of action of ascorbate in collagen assembly only occurred at the beginning of the 20th century [108]. More recently, two discoveries paved the way to the in vitro production of connective tissues. First, in 1972, by Switzer and Summer, was the demonstration that supplementation of fibroblast cell culture with sodium L-ascorbate stimulates the production of type I collagen [109]. Second, Hata and Senoo, in 1989, demonstrated that fibroblasts could deposit enough ECM, within a few days, to create a 3D stromal sheet [110].

### The Self-Assembly Technique (Also Known as Cell-Derived Matrices)

On the basis of this simple principle, a TE method called self-assembly was developed. In contrast to other techniques of TE, which use a scaffold, either synthetic or natural, on what seeded the cells (when cells are used), in the self-assembly technique, the cells build their own scaffold, providing an ECM microenvironment close to the native one. Mesenchymal cells, mainly fibroblasts, are seeded into culture dishes with ascorbate for a period of 3 to 5 weeks, depending on the amount of collagen secreted. An anchorage of paper, weighed down by metal supports, limits the tissue's contraction. A stromal sheet composed of many ECM proteins is then created. The sheets can be rolled around a mandrel to produce tubular tissues, whereas the production of flat tissues can be done by stacking several sheets and allowing their fusion. Such a scaffold can be seeded with epithelial or endothelial cells [111]. The technique was recently improved by replacing the stacking with a reseeding after two weeks to obtain a similar thickness [112]. The self-assembly method was used to produce several TE substitutes for grafting, or models for fundamental research into subjects including, but not limited to: blood vessels [113], skin [114] with its pathologic variations: hypertrophic scars [115], scleroderma [116], psoriasis [117], melanoma [118], recessive dystrophic epidermolysis bullosa [119], the cutaneous manifestation of amyotrophic lateral sclerosis [120]; cornea [121]; adipose tissue [122], urethra [123], studies on metal biodegradable ureteral stents [124], bladder [125] with bladder cancer [126] and ketamine-induced cystitis [127]; vagina [128] and the HIV infection of vagina [129]; and osseous tissue [130]. In all these substitutes, excellent 3D pseudo-capillary networks allowing quick reperfusion of the grafts can be formed by seeding endothelial cells [112,118,131–134].

### Original Method

The work published in 1998 by L'Heureux et al. [113] (Figure 3a) is the first successful assay to produce a completely biological TE vascular tube demonstrating sufficient mechanical properties to be grafted.

The self-assembled stroma, obtained using SMC and fibroblasts cultured for 30 days in Dubelcco's modification of Eagle's medium:Ham's F12 modified medium (3:1 ratio) with 10% fetal bovine serum and in the presence of 50 µg/mL ascorbate, were successively tightly rolled around a perforated PTFE tube. The first layer was a dermis sheet decellularized through dehydration. This layer was the scaffold for endothelialization of the luminal part. The second layer was the SMC-derived sheet. After one week of maturation, a third fibroblast-derived sheet was rolled around the construct. Eight weeks later, the tube was cannulated and filled with an endothelial cell solution to seed the luminal face, i.e., the acellular fibroblast-derived sheet. The TE blood vessel is mature enough to be used. The TE blood vessel produced by this technique presents histological (Masson's trichrome [113]) and mechanical (burst pressure [113,135]) properties close to native vessels, and a version without endothelial cells was grafted into dogs for seven days without any sign of deterioration [113]. Pharmacological studies were done using such TE blood vessels to measure contraction/relaxation potential [136]. These studies showed a contractile response of the construct. Mechanical and vasocontractile properties of the reconstructed tissue are also improved by mechanical stimulation induced by mechanical load [137] in a bioreactor. Similar results were obtained by using an engraved thermoplastic elastomer to guide the orientation of cells through surface topography [138]. Without including cell extraction and expansion, the whole process takes three months, rendering it not directly beneficial for the treatment of patients in an emergency. This is the major drawback of the technique.

Nevertheless, assays have been done to determine if decellularization can provide off-the-shelf vessels produced by the self-assembly method. This study showed similar in vitro mechanical properties after endothelialization of the substitute [139]. A clinical trial was conducted in 2009 [140]. Half of the patients were removed from the study for

various reasons (unrelated death, severe gastrointestinal bleeding, graft failure), and for the other half, the follow-up could be continued for 6 to 20 months.

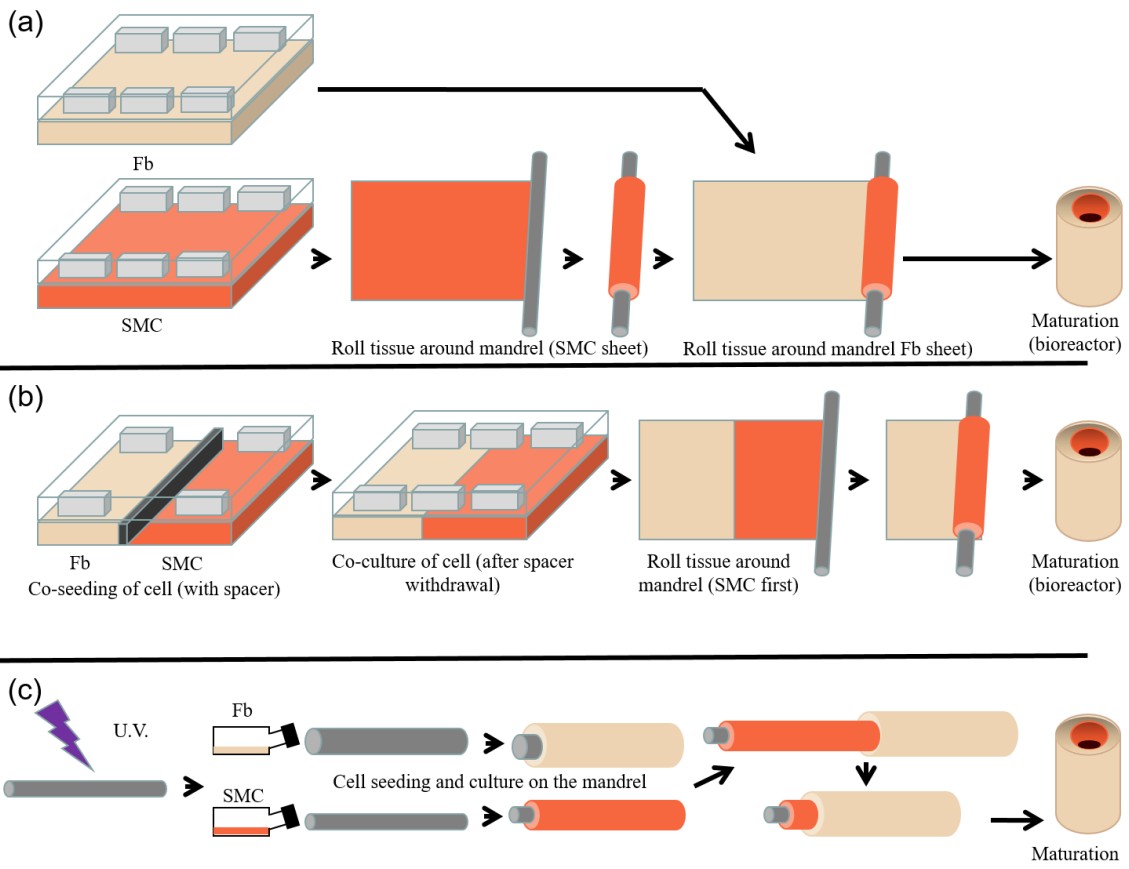

**Figure 3.** The self-assembly method and its improvements to produce vascular grafts. (**a**): Original method (L'Heureux et al.); (**b**): One-step method (Gauvin et al.); (**c**): Alternative method (Galbraith et al.).

Advanced Method

In the first method described above, successive sheets were rolled around a mandrel to produce the TE blood vessel. A modification of this technique was introduced to use only one sheet with both cell types SMCs and fibroblasts [141] (Figure 3b). Contrary to the original method, this technique only needs one step for rolling. SMCs and fibroblasts were seeded in the same culture dish, separated into two compartments by a custom-designed spacer (to prevent cell mix-up) removed 24h after the seeding step (to allow fusion of compartments). The sheet was then rolled, beginning from the SMC side. This technique results in the absence of separation between SMC and fibroblast sheets, because there is only one sheet. The tissue is histologically similar to the one produced using the original technique, with a thinner wall thickness (there were two sheets in the original technique and only one in the advanced method) but a greater ultimate tensile strength (UTS) and burst pressure. The decellularized fibroblast sheet rolled in the first position in the original method is unnecessary due to the ECM deposited in the SMC sheet, which allows endothelial cell colonization. Improvements have been made to the technique, especially in the recreation of a vasa vasorum to improve graft integration and vascularization through inosculation [131]. Recently, efforts have been made to harvest the required cells to produce the scaffold in a less invasive way. Adipose-derived stromal/stem cells (ASC) provide an up-and-coming source of cells for TE [142]. ASCs were used to produce a completely autologous TE blood vessel [143]. These constructs were compared with the ones produced using fibroblasts without a notable difference in wall thickness, failure load, burst pres-

sure or suture strength. These results indicated that ASCs could be a viable alternative to other cell sources. It should also be noted that ASCs have interesting proangiogenic and immunomodulatory properties [144]. The tissues produced using the self-assembly technique can also be used differently. Rather than rolling sheets, tissues can be cut and twisted to produce yarns to create human-derived textiles. The decellularization of the tissues allows their long-term preservation. These yarns were used to produce braided or twisted multifilaments, by knotting, crocheting, knitting, winding, or weaving tissues, which can recreate vascular prosthesis [145].

Alternative Method

The fusion between the sheet rolls is a crucial step of the self-assembly method to produce TE blood vessels. The ends of the rolls are especially subjected to infiltration by cell culture medium during maturation or blood after the graft, producing delamination with a complete loss of the mechanical and functional properties of the reconstructed substitutes. An alternative method has been developed to rule out these adverse events [146] (Figure 3c). In this work, polyethylene terephthalate glycol-modified (PETG) mandrels were treated with 30 min UV-C (254 nm) at five mW.cm$^{-2}$. Mandrels were seeded with SMCs or fibroblasts and cultivated for 21 to 28 days. To obtain multilayered TE blood vessels, tubular constructs produced using fibroblasts were stacked onto tubular constructs produced by SMCs, only by gliding constructs on each other. An additional period of 21 days was required to obtain a complete fusion of the stacked tissues. For the same wall thickness, the TE blood vessel produced by the alternative method showed better results in terms of UTS and burst pressure than the ones produced using the original technique.

*2.5. Cells Used for Vascular Tissue Engineering*

The production of tissue-engineered vascular graft can use many cell types to reconstruct the blood vessels' three layers. As reviews have been published on this subject, only a summary will be provided here [31]. Autologous cells, such as vascular or dermal fibroblasts, were used for the tunica adventitia. Alongside smooth muscle cells, which have a lower potential for cell expansion once differentiated, progenitor or stem-cell-derived SMC, from various tissues, such as bone marrow stem or mononuclear cells (BM-MSC and BM-MNC, respectively), but also muscle-derived stem cells, hair follicle stem cells, hematopoietic stem cells (HSC) or adipose-tissue-derived stem cells (e.g., ASC), were used. It should be noted that some of these cells can also provide cells for other layers, especially the endothelial cells (EC) for the tunica intima. EC from vascular origin can be used for this layer, or EC derived from various kinds of stem/progenitor cells [147], including endothelial progenitor cells (EPC). The group of Yamanaka3 was first developed induced pluripotent stem cells (iPS) [148]. They are somatic cells transduced with transcription factors allowing their reprogramming into embryonic-like stem cells. They can be derived into the cells required to produce all three vascular layers [149,150]. At present, these cells are promising, but their clinical use is questionable, especially for pediatric patients [151].

## 3. Urological Domain

Urological tissues can be affected by several pathologies requiring surgical reconstruction to restore the normal genitourinary function. To do so, biological material is needed in amounts that are dependent on the severity of the patient's condition. However, there is an imbalance between the growing demand for tissues, due to the ageing of the population, increasing incidence of chronic diseases and progress in surgical procedures, and the available supply, which is reduced by the health status of patients and regulatory concerns, often to the detriment of the patient. Due to the lack of available adequate tissues for these genitourinary reconstructions, efforts have been made to think outside the box. Surgeries face significant aesthetic, functional and anatomic challenges when they affect children, for whom the physical and psychological consequences will follow them throughout their lives. Due to the high rate of complications and recurrences [152], surgical

techniques have been evolving over the years. With the emergence of TE, there is hope for the treatment of patients with the most severe forms and those with recurrence, which require the most considerable amounts of tissue. Synthetic, natural, cellular or acellular biomaterials have been established to perform surgeries, but the resulting recurrence rate remains unacceptably high [153]. This part of the review will focus on the genitourinary aspects of tubular TE.

### 3.1. Anatomy

Ureters and the urethra are tubular structures that share the same leading role: expelling urine from the body. The kidneys are connected to the bladder with the ureters, which begin at the uretero–pelvic junction to enter the trigone's bladder. Its walls are composed of three primary tissue: inner mucosa, middle muscle layer and outer serosa (Figure 4a). Arbitrarily, ureters were divided into three parts: proximal, middle and distal parts, abdominal, pelvic and intramural segments.

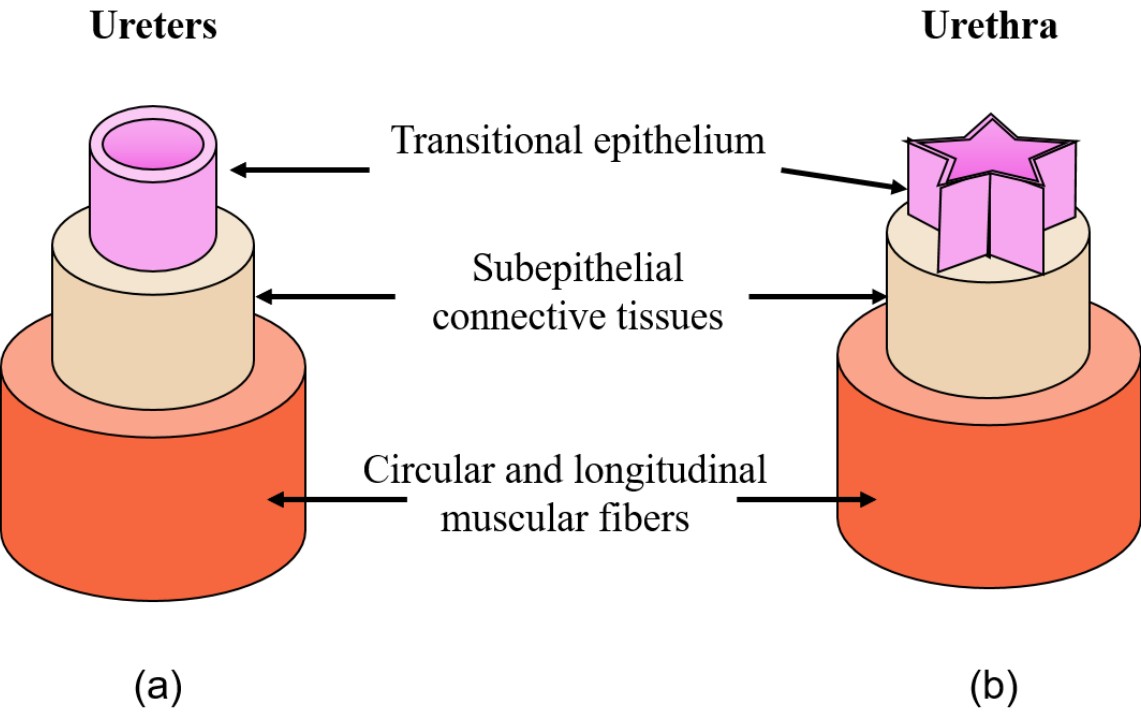

**Figure 4.** Schematized anatomy of urological conduits. (**a**): schema of the ureters' anatomy; (**b**): Schema of the anatomy of the urethra.

Once in the bladder, urine is transported outside the body through the urethra schematized in Figure 4b. Urine movement is facilitated by the peristaltic activity of the surrounding muscular layers. These muscle layers also compensate for the increase in pressure caused by the urine stream. The urethra has an additional feature: it allows for the progression of seminal fluids in men. Indeed, the human penis is the male organ for urination and copulation, composed of various tissues that are critical to maintaining its functionality. On the back of the penis, side by side, two dorsolateral corpora cavernosa provide erectile functionality. Indeed, those two cylindrical tubes become filled with blood during the erection phase to maintain the penis in its erect form. Indeed, the deep dorsal vein constriction prevents the outflow of blood.

The corpora cavernosa are surrounded by elastin fibers forming the tunica albuginea. On the ventral aspect of the penis, the corpus spongiosum is located, surrounding the urethra, from the prostate to the urethral meatus. The urological system is covered by a transitional epithelium whose primary role is to protect the tissues against urine. This is achieved through waterproof epithelial cell layers called the urothelium. This epithelium

varies from pseudostratified to pavimentous, depending on its location [154]. Three significant parts of the urethra have been characterized by these transitions and the surrounding tissues: prostatic, membranous and spongy urethra. The urothelium is laid on basal lamina and lamina propria, allowing innervation, vascularization and exchanges between the epithelial cells and the underlying tissues [155]. Due to the high degree of complexity of the urinary system, the correction of congenital/acquired malformation is still challenging.

### 3.2. Pathologies

This review will focus only on pathologies appearing in ureters or urethra, which may require the replacement of tissues using a bio-engineering tubular graft. The first focus will be on ureteral pathologies; these damaging urethra will be discussed in the second section.

### 3.2.1. Ureters

Due to its location, the most common causes of ureteral injury are iatrogenic, comprising 80% of the cases, while 20% are due to external trauma [156]. Some studies indicate that its incidence varies between 0.5% to 10% [157], while others claim lower rates from 0.3% to 1.5% [158]. However, studies agree that ureteral injury occurs most commonly during gynecologic surgeries, with an estimation of an incidence from 52% to 82% [156] of cases. A consensus indicates that injuries are due to the most challenging surgeries [157]. Indeed, ureters travel into the pelvis near the infundibulo-pelvic ligament and uterine artery and have a similar vascular structural appearance. This is even more challenging when a hostile environment is encountered due to repeated surgeries, inflammation, or neoplasm.

Additionally, unexpected congenital anomalies, such as ureteric duplication or retrocaval ureter, can increase the surgeries' complexity. Because injuries may not be apparent during the surgery, the symptoms encountered for these patients are mainly flank pain, ileus, hematuria, and prolonged high drain outputs with an increase in creatinine [157]. The failure to perceive symptoms can lead to severe side effects, including sepsis or renal function loss.

### 3.2.2. Urethra

The urethra is more subjected to external trauma, particularly for men, due to its length and location. Therefore, urethral pathologies can be divided into two major groups: congenital or acquired [159].

#### Congenital Urethral Anomalies

Hypospadias is the most common penile malformation, comprising 73.3% of all congenital penile anomalies: one in 250 newborn boys is affected [160,161]. This anomaly results from a defect of the tubularization and the fusion of the ventral urethral plate. Several studies have recently reported an increased prevalence of hypospadias in males [162]. It has been shown that 30% of births with hypospadias have a genetic cause, with a mutation in genes such as WT1 or AR [163]. Exposure to endocrine-disrupting chemicals or drugs has also been associated with a higher risk of hypospadias [164]. Affected patients are characterized by a position of the urethral meatus under the tip of the glans penis, close to its expected normal position, on the ventral face of the penis. The severity of the anomaly and the complexity of its correction are positively correlated with the position of the meatus compared to its normal position. The most severe cases show infertility in addition to psychological consequences. Epispadias is rarer, occurring in 1/117,000 boys [165]. This anomaly is also characterized by a misposition of the urethral meat, but on the opposite side of the penis. It can appear from the dorsal side of the penis to the pubic symphysis. The worst cases involve vesical exstrophy. In this case, the sphincter damage leads to incontinence and infertility. Other symptoms are also commonly found, such as reflux and urinary infection.

Chordee, defined as a curvature of the erect penis, can be acquired or congenital, but both may need concurrent urethral reconstruction depending on the severity. The first one is

generally due to fibrous tissue between the corpus cavernosum and the corpus spongiosum. In contrast, the congenital variant is due to the asymmetrical development of the corpus cavernosum. Its prevalence has been estimated to be between 0.5% and 20.3% [166].

Acquired Urethral Anomalies

Stenosis of the urethra, called stricture in the anterior urethra, is the most commonly acquired urethral anomaly, with an incidence of 0.6%, leading to more than 5000 hospital visits per year in the United States [167]. Stenosis appears when the urethral lumen becomes impeded by surrounding fibrotic tissues, leading to a narrowing or even a total obstruction of the conduit. The most common causes of stenosis are injury/trauma, such as motor or bicycle accidents, but it can also be induced by infection, instrumentation, lichen sclerosus or cancer. In addition to micturition difficulties, patients with stenosis have a high rate of urinary tract infections (41%) and incontinence (11%) [168].

Patients presenting recurrence or severe congenital and/or acquired anomalies of the ureters or urethra may need surgical reconstruction to re-establish a normal cosmetic appearance and normal function. The next section will only focus on surgeries to re-establish penile function.

### 3.3. Current Therapeutic Solutions

Current treatments allow for the good management of patients with minor pathologies of the genitourinary system. However, it is more difficult to properly treat patients, with a positive long-term outcome, for more severe diseases. In this section, we will discuss the current therapeutic solutions and gold-standards for ureteral and urethral reconstructions.

### 3.3.1. Surgical Procedures for Ureters Reconstruction/Repair

Depending on the diagnosed pathology, surgery may be performed. Various surgical procedures may be performed, depending on the pathology stage and evolution: endoscopic surgery, partial ablation by segmental resection or total ablation of the ureter by radical nephroureterectomy. Patients presenting short-length ureteric disorder are treated with end-to-end anastomosis due to its better results [169]. For the long-segment reconstructions, autologous bowel tissues can be used [170]. If the defect occurs in the distal portion of the ureter, the bladder could be reconfigured to allow direct ureteral reimplantation (Boari flap). However, the type of bowel segment used may cause complications, including metabolic imbalance, malabsorption of vitamins, lithiasis and infections [171] (Table 2). Multiple procedures are frequently needed over time due to stenosis or strictures, which may require another surgery and possibly a nephrectomy [170].

### 3.3.2. Surgical Procedures for Urethral Reconstruction/Repair

As the anatomical variation, etiology, localization, and severity of cases can lead to unpredictable functional and esthetic outcomes; no standard procedures can be established for all patients. Depending on the severity of the penile anomalies, few tissues can be used to replace or repair. Skin grafts, including genital and extra-genital skin flaps, tunica vaginalis, the tissue around a testicle [172,173], and lingual or buccal mucosa have been tested [174–181]. Described for the first time for urethral replacement by the Russian surgeon Kirill Sapezkho in 1890, oral mucosa remains the gold standard [182]. However, these approaches could also result in many complications, such as re-stenosis, pain, numbness, submucosal scars, dry mouth, lesions, neuro-sensory defects, discomfort, difficulty to open the mouth, risk of infection, and a limited amount of tissue that can be harvested, which is problematic for extended defects (Table 2) [183–188].

Current treatments for severe pathologies of ureters and urethra are mainly limited by the specific anatomical characteristics and site of injury, and lack of native tissue for replacement/repair [170]. Furthermore, many long-term complications have been documented following these interventions. Therefore, alternative graft materials are

needed. The emergence of TE may bring solutions to circumvent gold-standard-related complications and restore the functionality of the diseased organs.

### 3.4. Advanced Solutions: Tissue Engineering

Innovative strategies have been required to overcome the lack of available tissues for the genitourinary reconstructions. Indeed, the use of TE grafts may prevent complications, thus reducing the health costs, in addition to improving the quality of life of the patients. To replace genitourinary tissues, the biomaterials must have good mechanical properties, be biocompatible, and allow cell–cell communication, cell adhesion, differentiation, and migration, to respond to the tissue-specific functional demands. It must also be rapidly vascularized, be bioresorbable or biodegradable in a reasonable time, and similar to the scar tissue's regeneration kinetics. Finally, without generating toxic degradation products, it must allow tissue regeneration without causing an immune response after implantation [189,190].

### 3.4.1. Synthetic Biomaterials

Synthetic scaffolds have been tested for both ureters and urethra reconstruction. Two groups can be distinguished: degradable and non-degradable scaffolds. First made for blood vessel reconstruction [191] then adapted for genitourinary reconstruction, PGA, PLA, PTFE, and more have been tested [192–194]. Positive first opinions on these biomaterials arose due to their low cost and tunable mechanical properties, the fast-reproducible results with a low risk of contaminants, their availability, and the possible incorporation of substance growth factors of this 3D biocompatible organs [192–195]. Despite the previous advantages of these biomaterials, pre-clinical studies have been limited by the observed complications. Among these, the long-term effects of material degradation products are poorly understood and may exacerbate the inflammatory reactions (Table 2) [194].

Furthermore, the environment they create is mostly inadequate for the adequate differentiation and organization of epithelial cells. This epithelial defect can lead to non-functioning or poorly functioning tissue, allowing the passage of urine through the tissue and, therefore, forming strictures through inflammation [192,193]. Due to its mechanical characteristics and permeability, studies have shown that materials such as PTFE scaffolds are not suitable for ureteral/urethral reconstruction [191].

Therefore, to mimic soft tubular tissues, more elaborate structures emerged with electrospinning, offering different shapes, porosities and fiber arrangements [196]. However, the functionality of these materials still must be characterized in vivo [197]. Natural scaffolds have been tested as an alternative to the inconvenient synthetic biomaterials.

### 3.4.2. Natural Biomaterials

Alternative scaffolds, using polymers such as collagen type I [198] or silk fibroin derived from Bombyx mori cocoons [199], were tested. Collagen tubular structures have been reconstructed and grafted with a lower success rate than gold-standard procedures [200]. Several improvements have recently been performed using these hydrogels [201,202], but further studies remain necessary to evaluate the potential of Collagen type I for urethral reconstruction. Collagen type I scaffolds need to be crosslinked to reduce enzymatic degradation speed and enhance their mechanical properties, but this could cause them to lose their natural characteristics. Silk-fibroin-made scaffolds are characterized by their excellent physical characteristics [192] and induce few immunogenic and inflammatory responses, suggesting better biocompatibility than conventional urological biomaterials [203]. However, despite its exciting mechanical properties, this biomaterial is composed of only two elements: silk fibroin (75%) and sericin (25%), which are far from the native tissue and could impact the differentiation of the urothelium. Furthermore, the biodegradability is very long or even non-existent [204,205].

### 3.4.3. Decellularized Matrices

The most studied natural biomaterials are still animal matrices or cadaveric organs, which have been enzymatically, physically and/or chemically decellularized. This strategy ideally allows for the preservation of an acellular matrix whose micro- and macro-architecture, mechanical properties, and biochemical environment are identical to native tissues. The best known are the Small Intestinal submucosa (SIS) and the Bladder Acellular Matrix (BAM) [206]. The SIS is mainly formed from the submucosa of the small intestine of pigs, where the mucous, serous and muscular layers are mechanically removed to leave a 0.1 mm collagen-rich membrane [207,208]. Pilot studies and in vivo animal experiments have highlighted the benefits of this kind of graft: its complete degradation in 4 to 8 weeks and the advantage of stretching under force combined with its low tendency to tear [209].

Additionally, decellularized matrices for allow rapid cell growth and significant angiogenesis, comparable to the skin and mucosal grafts, probably because cell-binding molecules and specific growth factors are retained [210–212]. Despite the natural biomaterials' real advantages, residues of biological elements in these tissues, such as DNA or prion, represent an immune risk for patients (Table 2) [213]. To avoid this, more stringent decellularization protocols can be implemented. However, the decellularization process can alter the resulting tissue's biomechanical functionality, which could impair its use as a graft [214]. These tissues' vascularization remains very limited and could induce ischemia and necrosis of the implanted biomaterial and, therefore, graft failure [215].

Furthermore, urothelial regeneration in acellular biomaterials such as SIS or BAM is limited to 0.5 cm horizontally, compromising success in more complex cases such as long strictures [216]. In these cases, this biomaterial does not allow for the maturation of a tissue that will fulfil its waterproof function as soon as the graft is performed. A urinary diversion must be carried out to allow the tissue to mature and become functional in the prevention of extravasation of urine and risk of inflammation (Table 2) [217]. Finally, clinical experiences showed that, in addition to frequent infections found in these patients, the donor's age and the intestine area from which the SIS is derived impact the regenerative potential. These reasons have prevented this biomaterial from being considered as an ideal alternative to genitourinary repairs [210,218,219].

Despite the many studies that have been reported, the resulting recurrence rate remains unacceptably high [153]. To better meet clinical performance requirements, natural and synthetic materials have been combined to produce hybrid biomaterials.

### 3.4.4. Hybrid and New Generation of Biomaterials

Hybrid scaffolds combine natural and synthetic pros with improvements to their global quality and a reduction in the related complications. Collagen scaffolds have been reinforced with synthetic polymers to provide better mechanical characteristics while maintaining the biocompatibility [220]. Hybrid biomaterials showed a significant improvement compared to the collagen-alone graft, which is promising [221]. As expected, it has been shown that total-biodegradable-collagen-based hybrid grafts are superior to semi-biodegradable hybrids in terms of elastin deposition and muscle/urothelial cells' arrangement [222].

With the TE field's emergence, a new generation of biomaterials appeared, which also shows the increasing necessity of graft biomaterials. Thus, "Intelligent" biomaterials have recently appeared, which reversibly respond to temperature, ionic strength, pH or light [192,223,224]. Regarding synthetic biomaterials, these scaffolds can be loaded with the ECM or growth factor components to enhance cell adhesion, communication, differentiation, and migration. External stimuli such as pH or temperature can trigger the degradation of the scaffold [225]. These "advanced" biomaterials were first elaborated for the administration of drugs and medical devices [192], but can be adapted for urological use. Shape-memory polymers were also initially studied in vascular and bone tissue engineering but can be applied in the genitourinary domain [226,227]. This kind of biomaterial can extend, in a brief form, and return to its original state upon exposure to a stimulus [228].

It can be adapted to the urethra to stretch during erection and return to its original shape during detumescence. Even if the latest technologies, such as "intelligent scaffold", may be promising, there is still much work to be done. Precise control of the scaffold architecture and composition may allow for the control of its mechanical and physiological properties.

### 3.4.5. Bioprinting for Urologic Applications

A new way of fabricating 3D-tissues gives promising results, even if improvements are required. Indeed, the recent introduction of 3D-printing has been used to build substitutes with gradients, e.g., a gradient of biologically active factors, for the regeneration of cartilage and bones [229]. It has also been successfully applied to mimic several heterogeneous tissues like skin [230], vasculature [231], urethra [232] and bladder [233]. Indeed, this technology may be applied to the genitourinary field to produce ureters and urethra. It could allow for the high-resolution control of the substitute's microarchitecture, including the simultaneous printing living cells inside synthetic/natural biomaterials, with dimensions adapted to each patient. As a proof of concept, a tubular urethra has been produced for the first time using primary urothelial cells and SMCs within a combination of the fibrin-based hydrogel and polycaprolactone/poly(L-lactide-co-ε-caprolactone) (PLCL) blends as scaffolding biomaterial [232]. They showed that a tubular construct with mechanical properties similar to native tissues could be obtained. However, the small size of printable scaffolds, limited bio-inks available, and its slow processing time are disadvantages of this technology [234].

Furthermore, complex tissues are hard to produce due to their micro- and macro-architectures [234]. Finally, the impression of complex substitutes may not be widespread, due to their high cost and time-consuming nature. Further refinement of the techniques is required, but this technique is in its infancy.

### 3.5. Self-Assembly Technique

Using autologous cells from the patient to produce their scaffold seems optimal to avoid immunologic reactions with exogenous biomaterials. As described previously, the self-assembly method, which relies on this principle, has been successfully applied to produce vascular tubular grafts. Therefore, it has been adapted to genitourinary needs (Figure 5).

The first step demonstrates that human dermal fibroblasts ECM and porcine urothelial cells can be used to produce a genitourinary tubular tissue graft resistant to suturing and high internal pressures [123]. The mechanical properties of the substitute are more than sufficient to fulfill urethral functions. Using a custom-made bioreactor allowed for the adhesion, proliferation and differentiation of urothelial cells inside the tube. After that, the subsequent study used human urothelial cells instead of porcine ones and went further to characterize the urothelium's maturation, especially the permeability and the histology. A dynamic culture condition enhanced the tissue maturation, allowing its full functionality in vitro [235]. Indeed, it is believed that the mechanical stimuli applied by the flow of the cell culture medium can mimic the flow of urine, challenging the cells to improve their maturation to fulfill their natural function. The acquisition of well-localized vital proteins, such as Zonula occludens (ZO)-1, in the lateral junctions of umbrella cells, and uroplakin (UP)-2 at the apical surface of the urothelium, allows us to believe that the substitute could fulfill its waterproof function. The latter has been confirmed with a permeability test using Franz-type diffusion cells. Indeed, the substitute cultured for 14 days under dynamic conditions showed a very low permeability, equivalent to a native porcine urethra. The presence of mature umbrella cells has also been characterized by electron microscopy. Plaques of UP were visible with microvilli and microridges at the apical surface, and discoid and fusiform vesicles in the umbrella cells near the lumen.

Rapid vascularization is essential for the success of the transplant. The same human tubular substitute can also be successfully endothelialized if needed, and this was per-

formed [132]. After subcutaneous grafting of the constructs into mice, the presence of mouse red blood cells inside the human endothelial network demonstrated its functionality.

**Table 2.** Strengths and weaknesses of the transplants or biomaterials used for both ureteral and urethral reconstruction.

| Type of Biomaterial | Tissue | Strengths | Weaknesses | Ref. |
|---|---|---|---|---|
| Autologous transplants | Tunica vaginalis, genital skin flap, extra-genital graft | - High success rate<br>- Easily accessible, free of hair (prevents lithiasis formation), elastic<br>- Present in abundance with a vascularized thick mucous layer and submucosa | - Stricture and stenose formation, sometimes graft failure<br>- Quantity-limited and risk of infection<br>- Non-functional tissue | [180,181,183–188] |
| | Lingual mucosa | In addition to the strengths of the other autologous transplants:<br>- Tolerant to liquids<br>- Avoid aesthetic disadvantages found during the removal of penile skin | In addition to the weaknesses of the other autologous transplants:<br>- Sampling induces side effects in 80% of patients<br>- Pain, numbness, submucosal scards, dry mouth, lesions<br>- Neuro-sensory defects, discomfort<br>- Disorders of the opening of the mouth | |
| | Bowel tissues | - Easily accessible, elastic<br>- Present in abundance with a vascularized thick mucous layer and submucosa | - Metabolic imbalance, malabsorption of vitamins<br>- Cholelithiasis, nephrolithiasis and infections | [170,171] |
| Synthetic | PLA, PGA, PLGA, PCL | - 3D biocompatible organs<br>- Low cost and malleable, mechanical properties<br>- Fast and reproducible results with a low risk of contaminants<br>- Available at all times<br>- Allows the possibility of incorporating substances (e.g., growth factors) | - Long-term effects of the degradation production of the matrix still unknown<br>- Inadequate environment for the differentiation and organization of epithelial cells, leading to non-functional graft | [192–195] |
| Natural | SIS, BAM | - Rapid cell growth<br>- Significant angiogenesis, comparable to the skin and mucous membrane grafts<br>- Fully biodegrades in four–eight weeks<br>- Able to stretch under force and low tendency to tear | - Immune risk (DNA, prion)<br>- Vascularization minimal (necrosis)<br>- Regeneration limited to 0,5 cm<br>- Requires "urinary diversion"<br>- Unfavorable clinical experience, infections being the most significant limitation<br>- Age of the donor and the area of the matrix harvesting impact the regenerative potential | [206–213,215–217] |
| | Silk fibroin | - Excellent physical characteristics (elasticity and tear resistance)<br>- Better biocompatibility compared to conventional urological biomaterials (less immunogenic and inflammatory responses) | - Biomaterial composed of only two elements: silk fibroin (75%) and sericin (25%)<br>- Very long or non-existent biodegradability | [192,199,203–205] |

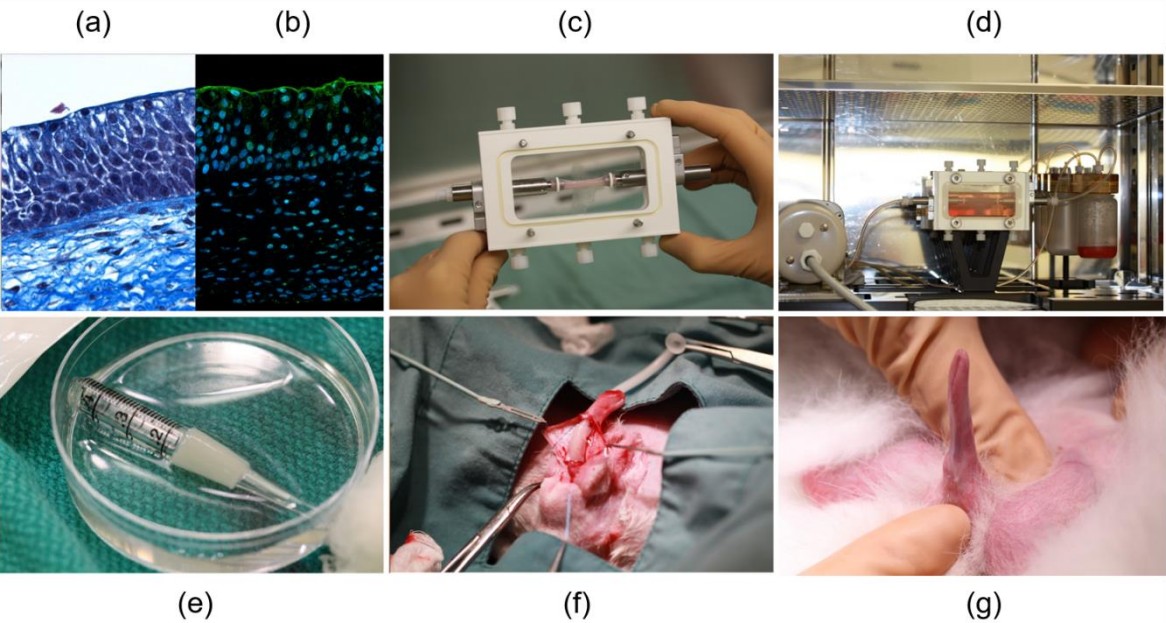

**Figure 5.** Self-assembly tissue-engineered urethra segment: from the lab to the animal. The upper panels represent successively (**a**) a Masson's trichrome staining of a slice of TE urethra segment; (**b**) immunofluorescence against uroplakin-2; (**c**) TE tubular urethra in a cassette of the LumGen bioreactor, and finally, (**d**) a TE urethra segment in the bioreactor to simulate urine flow. The lower panels represent the implantation, into rabbits, of a 1.5cm segment of TE urethra produced by the self-assembly method after maturation in the LumGen bioreactor. This segment was reconstructed without epithelium to avoid an immune reaction. The rabbit epithelium can recolonize short segments after grafting; the first panel (**e**) shows the TE urethra segment; the second panel (**f**), the implantation of the TE urethra graft to replace a portion of the rabbit's urethra; and finally, the third (**g**), the erection test on rabbit penis after the implantation of the TE urethra segment produced by the self-assembly method [236].

The previous results were obtained using dermal fibroblasts to produce the scaffold on which the urothelial cells were seeded. However, even better results could be expected when using organ-specific cells to produce the stromal part of the tube. The relationship between the lamina propria and the urothelium is critical for adequate differentiation of the epithelium [237] Using the self-assembly method, this was demonstrated, in a corneal reconstruction model using a cross-combination of dermal fibroblasts or keratocytes with corneal epithelial cells or keratinocytes. With an inadequate combination of the stromal and epithelial cells, the epithelium's differentiation and functionality were reduced [238]. Similar experiments were done using cells derived from porcine biopsies. Dermal fibroblasts, bladder mesenchymal cells, or a mix of both cells, produced the stromal compartment on which urothelial cells were seeded. Demonstrating the impact of organ-specific cells in the reconstruction of the lamina propria, bladder mesenchymal cells improve the normal urothelial differentiation, avoiding the incorrect localization of keratin 14 [239]. However, in the condition used in this study, bladder mesenchymal cells deposited much less collagen in the stroma than dermal fibroblasts, and therefore weakened the mechanical strength of the substitute. Subsequent studies comparing native tissues with collagen hydrogels as scaffolds showed that soluble mediators seemed more critical to this effect than the ECM composition/organization [240]. Therefore, further studies are required to improve the mechanical resistance of the substitute while retaining the advantages mediated by the organ-specific cells' secretome.

The main advantage of the self-assembly method in the production of tubular genitourinary grafts is autologous organ-specific cells. This reduces the risk of inflammatory reactions, because cells and proteins are derived from the patient's transplant rejection by favoring rapid graft integration into the host. Furthermore, the produced substitute is functional as soon as the implantation is performed, avoiding urine leaks. This fully

biological tubular genitourinary substitute, with a living scaffold, and epithelium that retains physiological cell–cell and cell–matrix interactions, should grow with the child when implanted in a pediatric patient. This potential is also supported by the presence of urothelial stem/progenitor cells [241]. However, as a drawback, the production of this highly personalized substitute takes 2 to 3 months. It requires specialized human resources, resulting in high cost, even if efforts have been made to reduce the time and money required to reconstruct the tissues [112,242].

Nevertheless, virtually all urologic cases, even the most serious, are not urgent or potentially fatal, as temporary solutions can preserve the patient, allowing time to produce the substitute. Concerns about the price are legitimate, but further preclinical and clinical studies should determine whether this type of substitute prevents recurrence, as was the case with the treatment of severely burnt grafted patients using the self-assembled skin [243]. These recurrences after urethral repair/replacement currently represent a high burden on health-care resources.

### 3.6. Cells Used for Urologic Tissue Engineering

Various cells have been used to reconstruct the ureters and the urethra. They are described in two recent reviews [244,245], and we summarize them here. Urologic tissue can be divided into three layers, even if lamina propria is omitted, despite its crucial role in urothelial and detrusor differentiation/maturation. The Detrusor muscle layer can be reconstructed using adult cells, such as smooth muscle from ureter, bladder, or corpus cavernosum biopsies, but also smooth muscle-differentiated stem cells such as urine-derived stem cells (USC or UDSC), adipose-derived stem cells (ASC, sometimes written as ADSC), mesenchymal stem cells (MSC). The Lamina propria layer can be engineered from adult cells such as dermal fibroblasts, oral mucosa fibroblasts, or bladder mesenchymal cells (BMC) from urologic tissue biopsies. Stem cells, such as MSC or ASC, could also be used, due to their potential for differentiation into secretory-type fibroblasts. Finally, the urothelium layer was reconstructed using adult cells, such as cells from bladder washing, urothelial cells from urologic tissue biopsy, oral mucosa keratinocytes, or epidermis keratinocytes. Stem cells such as ASC were also used. It can be noted that adult cell populations contain stem cells, which can be expanded in cell culture. Their potential could be limited, especially when the patient is diseased. In this case, induced pluripotent stem cells (iPS) [246,247] or direct reprogramming of cells [248], derived from blood cells, fibroblasts, or UDSC, could be used in the future of the reserve described above for cell sources in vascular TE.

### 3.7. Perspectives and Challenges

Instead of adding a synthetic biomaterial to a natural one to improve its mechanical characteristics, an imitation of solutions found by the evolution is always a convenient choice. This led to the testing of a particularly ingenious solution by producing a urethral tissue closer to the natural one [249]. Indeed, a star-shaped scaffold was produced, mimicking the folds in the lumen that we naturally find in the urethra. The tube's radial elasticity was successfully improved, increasing the burst pressure from $52 \pm 21$ mmHg in a classic circular tube to $132 \pm 22$ mmHg in a star-shaped tube. In comparison, pig urethras presented a burst pressure of $274 \pm 93$ mmHg. However, it should be noted that native tissue has additional muscle tissue surrounding the urethra, which may be added in further studies.

As previously described, a new generation of TE blood vessels, using human dermal fibroblasts with or without SMCs, was produced without the need to roll stromal sheets around a mandrel [146]. Indeed, a human tubular scaffold could be obtained by seeding the dermal fibroblasts directly on the mandrel. The burst pressure obtained in both cases seemed similar to the native artery used as control. This technique could be adapted to the genitourinary domain by seeding the bladder mesenchymal cells directly on the mandrel to produce a tubular substitute.

As previously mentioned, the urethra is a multisegment tissue with variation in the epithelium organization, depending on the location. As was done for the self-assembled blood vessels, two compartments can be created using a spacer during the cell seeding [141]. Nevertheless, in the case of urethral reconstruction, instead of creating, as in the vascular tissue, successive SMC layers followed by a series of fibroblast layers by rolling the stroma around the mandrel in the parallel orientation of the separation line, the tissue could be rolled in the perpendicular orientation of the separation line to create adjacent stromal layers (dermal vs. bladder stromal cells). This technique could allow the recreation of the epithelial transition zones, as found in the native urethra. Indeed, the underlying molecular and cellular pathway explaining the transition from a pseudo-stratified epithelium to a squamous epithelium in the distal part of the urethra remains unknown [250]. The use of stromal and epithelial cells originating from the bladder or different urethral segments would allow several combinations that could be used to investigate the signals required to obtain two distinct zones. After that, the technique could be applied to produce a tubular scaffold with a pseudo-stratified and squamous transition. The squamous part in the distal urothelium plays an essential role in the prevention of urinary infection [251].

## 4. Conclusions

Many factors are known to significantly increase the risk of developing pathologies. These include the aging of the population, the globalization of the Western way of life, and the spread of endocrine disruptors in the environment. Patients developing congenital or acquired anomalies may require the replacement/repair of organs and tissues. This is particularly true in the vascular and urological field. Due to the lack of adequate autologous tissue available for corrective surgeries, alternative solutions have emerged. Tissue engineering is one of the scientific advances which could represent an alternative in the most severe cases or for patients suffering from recurrences. This recent field relies mainly on the use of natural or synthetic biomaterials to produce tissues and organs. However, despite its numerous advantages, the use of exogenous biomaterials can generate some known side-effects. Tissue engineering has been revolutionized by the appearance of several new technologies, such as 3D or 4D bioprinting and the self-assembly technique. This latter method allows the production of 3D substitutes without the need for exogenous materials. As it is improved year after year, the self-assembly method, implemented for the reconstruction of blood vessels and skin, has extended its potential for use in several organs, including the ureter and urethra. Furthermore, this technique has also been derived in innovative human 3D research models. Much remains to be done, especially concerning the determination of the long-term security of patients, but these alternatives could improve the patient's quality of life.

**Author Contributions:** All authors contribute to write, review and edit of the text. All authors have read and agreed to the published version of the manuscript.

**Funding:** This study was supported by a Canadian Urological Association Scholarship (S.B.), a Canadian Institutes of Health Research Grant (#258229) (S.B.), the 'Fonds de Recherche du Québec—Santé (FRQS) (C.C.).

**Institutional Review Board Statement:** Not applicable.

**Informed Consent Statement:** Not applicable.

**Data Availability Statement:** Not applicable.

**Acknowledgments:** The project is supported by the Quebec Cell, Tissue and Gene Therapy Network—ThéCell (a thematic network supported by the Fonds de recherche du Québec–Santé).

**Conflicts of Interest:** The authors declare no conflict of interest.

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
