# Peer review of "Reconstruction of Vascular and Urologic Tubular Grafts by Tissue Engineering"

_processes, doi:10.3390/pr9030513_

Round 1

Reviewer 1 Report

Summary: The authors of this review paper offer an up-to-date overview of the tissue engineering strategies for the vascular, ureteral, and urethral reconstructions. Likewise, they put emphasis on the self-assembly methods to obtain tissue/organ substitute without the use of exogenous material, but with the patient’s cells producing their own scaffold.

The authors should consider the following points:

1. For a better visual impact to the readers I advise the authors to provide a table denoting the different types of biomaterials used for obtaining tissue-engineered vascular grafts, encompassing the corresponding references.

2. Table 1 must be redrawn since it appears cut off (the lateral right side).

3. Minor concerns:

Line 512 – it is written “autologous bowel tissues can used”. Do you mean “autologous bowel tissues can be used”?

Line 579 – it is written “preserving an acellular matrix, which micro- and macro-architecture”. Do you mean “preserving an acellular matrix with micro- and macro-architecture”?

Reviewer 2 Report

The manuscript entitled 'Reconstruction of vascular and urologic tubular grafts by tissue engineering' by Caneparo et.al describes the methods of tissue reconstruction focused on vascular and urological grafts. Overall manuscript is well written, with brief background of tissue engineering authors have discussed the current techniques and associated challenges. The English language used by the authors is appropriate and the review is properly cited.

In light of this, the manuscript may be considered for publication in its current form.

Reviewer 3 Report

Dear Author,

the review focused on the reconstruction by tissue enginneering of vascular and urological grafts is well described and detailed. 

In my opinion the author should improve the description of sources of autologous cells usable for tissue construction.

Moreover the table 1 is not visible on the document. Please send the correct version. 

Best regards

Round 2

Reviewer 1 Report

The authors considered my comments. I only suggest to them to make the next correction into the final manuscript:

1) Table 1. (Acellular matrix): “Mechanical properties (depending of the protocol used), DNA and antigen residues” corrected to “Mechanical properties (depending on the protocol used), DNA and antigen residues”

2) “The group of Yamanaka3 (?) has first developed” (line 419)